# Proton Therapy for Advanced Juvenile Nasopharyngeal Angiofibroma

**DOI:** 10.3390/cancers15205022

**Published:** 2023-10-17

**Authors:** Line Hoeltgen, Thomas Tessonnier, Eva Meixner, Philipp Hoegen, Ji-Young Kim, Maximilian Deng, Katharina Seidensaal, Thomas Held, Klaus Herfarth, Juergen Debus, Semi Harrabi

**Affiliations:** 1Department of Radiation Oncology, Heidelberg University Hospital, 69120 Heidelberg, Germanysemi.harrabi@med.uni-heidelberg.de (S.H.); 2Heidelberg Ion-Beam Therapy Center (HIT), Department of Radiation Oncology, Heidelberg University Hospital, 69120 Heidelberg, Germany; thomas.tessonnier@med.uni-heidelberg.de; 3National Center for Tumor Diseases (NCT), Heidelberg University Hospital, 69120 Heidelberg, Germany; 4Clinical Cooperation Unit Radiation Oncology, German Cancer Research Center (DKFZ), 69120 Heidelberg, Germany; 5Heidelberg Institute of Radiation Oncology (HIRO), 69120 Heidelberg, Germany; 6German Cancer Consortium (DKTK), Partner Site, 69120 Heidelberg, Germany

**Keywords:** juvenile nasopharyngea angiofibroma (JNA), proton radiotherapy, radiotherapy, NTCP, late effects, pediatry, secondary cancers, neurocognitive impairment

## Abstract

**Simple Summary:**

Advanced juvenile nasopharyngeal angiofibroma (JNA) can present a considerable therapeutic challenge. Photon radiotherapy has been recognized as an effective treatment approach for recurrent or inoperable advanced JNA in an additive or definitive setting. However, due to the benign nature of JNA and the young age of the affected patients, concerns about long-term radiation-induced morbidity have led to a certain hesitance towards this treatment modality. Proton therapy (PRT) presents a promising alternative for treating benign conditions in young patients, yet there remains a gap in the literature regarding its application to JNA. We retrospectively examined the outcomes of patients with JNA who underwent PRT in our center and evaluated the dosimetric advantages of PRT over conformal radiotherapy. This study is the first report investigating PRT for advanced JNA, highlighting PRT’s safety and effectiveness as a viable therapeutic option. Dosimetric and complication risk evaluations suggest that PRT holds the potential to reduce long-term radiation-induced complications, including the development of secondary neoplasms or decline in neurocognitive function.

**Abstract:**

Purpose: To provide the first report on proton radiotherapy (PRT) in the management of advanced nasopharyngeal angiofibroma (JNA) and evaluate potential benefits compared to conformal photon therapy (XRT). Methods: We retrospectively reviewed 10 consecutive patients undergoing PRT for advanced JNA in a definitive or postoperative setting with a relative biological effectiveness weighted dose of 45 Gy in 25 fractions between 2012 and 2022 at the Heidelberg Ion Beam Therapy Center. Furthermore, dosimetric comparisons and risk estimations for short- and long-term radiation-induced complications between PRT plans and helical XRT plans were conducted. Results: PRT was well tolerated, with only low-grade acute toxicities (CTCAE I–II) being reported. The local control rate was 100% after a median follow-up of 27.0 (interquartile range 13.3–58.0) months. PRT resulted in considerable tumor shrinkage, leading to complete remission in five patients and bearing the potential to provide partial or complete symptom relief. Favorable dosimetric outcomes in critical brain substructures by the use of PRT translated into reduced estimated risks for neurocognitive impairment and radiation-induced CNS malignancies compared to XRT. Conclusions: PRT is an effective treatment option for advanced JNA with minimal acute morbidity and the potential for reduced radiation-induced long-term complications.

## 1. Introduction

Juvenile nasopharyngeal angiofibroma (JNA) is a rare benign condition accounting for <1% of all head and neck tumors and occurring almost exclusively in young men [1]. The exact origin site of JNA remains unclear; however, most authors consider JNA to originate from the sphenopalatine foramen or the pterygoid canal [2]. Due to their locally aggressive nature, they have the potential to infiltrate and damage surrounding structures, for instance, the infratemporal fossa or the skull base, with intracranial spread in advanced stages [3]. Indeed, intracranial tumor extent is reported to occur in up to 37.5% of patients [4].

While common symptoms generally comprise nasal obstruction or epistaxis, advanced JNA can thus present with facial swelling, trismus, and visual or other neurological disturbances [5,6,7]. Diagnosis primarily relies on clinical and imaging characteristics [5,8]. A biopsy is not recommended routinely as JNA are highly vasculated tumors that can occasion severe hemorrhage with minor manipulation, not least due to their intratumoral vessels being deficient of a proper muscular layer and their stroma lacking elastic fibers [5,9]. 

Among the multiple treatment options discussed in the literature, external-beam radiotherapy and surgery are pointed out to be the most common and most effective modalities [7,10,11]. Whenever gross tumor resection can be achieved, the preferred treatment is considered to be surgical resection, regularly combined with preoperative embolization to reduce significant intraoperative haemorrhage and related complications [5,12]. However, the optimal approach to manage advanced stages remains controversial [13]. In the case of advanced stages with for example skull base involvement or intracranial extension, resection is often associated with high morbidity or bears the risk to be incomplete with potential subsequent recurrence of residual disease [8]. This substantiates the consideration of radiotherapy for recurrent or inoperable advanced JNA, either in an additive or definitive setting [7,14]. 

Still, a major concern regarding radiotherapy for a benign condition in young patients consists of long-term radiation-induced sequelae due to incidental dose deposition in normal tissue [7,15]. Due to the localization and tumor extent of JNA in advanced stages, it may be challenging to reduce radiation dose deposition in surrounding critical structures including structures of the central nervous system (CNS) or endocrinological structures. However, it is imperative to preserve long-term quality of life following radiotherapy by reducing the risk of radiation-induced long-term complications such as neurocognitive impairment, hormonal deficiency or also secondary neoplasms. 

While enhanced sparing of surrounding normal tissue has been suggested in individual cases through the use of stereotactic radiotherapy, caution should be exercised for this treatment modality when it comes to larger treatment volumes, which are commonly encountered in advanced JNA [7,16,17,18]. 

In comparison to contemporary conformal photon radiotherapy (XRT), such as intensity-modulated radiation therapy (IMRT), volumetric modulated arc therapy (VMAT), or helical therapy, the biophysical properties of proton radiotherapy (PRT) with low dose absorption on tissue entry and a nearly complete dose absorption in the Bragg-peak offer a higher degree of conformity with a steep dose gradient and a reduction in integral dose to normal tissue [19]. The benefits of PRT have been reported for various pediatric conditions in several planning treatment and clinical studies [20].

To the best of our knowledge, there have been no reports documenting the utilization of PRT for the treatment of advanced JNA. In this study, we report our experience with PRT in managing advanced JNA. In a second step, we further conduct risk assessments for long-term complications through plan comparisons with helical XRT.

## 2. Materials and Methods

### 2.1. Patient Selection

In this single-institution retrospective analysis, we analyzed all consecutive patients having received PRT for JNA in a definitive or postoperative setting between 2012 and 2022 at the Heidelberg Ion Beam Therapy Center. The analysis was approved by the Heidelberg University Ethics Review Board.

Information was obtained from the patients’ medical and radiation therapy records. Data collected included patient demographics, tumor characteristics, radiotherapy parameters, as well as symptomatic and tumor response.

### 2.2. Proton Treatment Planning and Delivery

A customized thermoplastic mask was used for immobilization. For treatment planning, contrast-enhanced magnetic resonance imaging (MRI) and a computed tomography (CT) scan were acquired. Gross tumor volume (GTV) was defined as all visible macroscopic tumor volume on contrast-enhanced MRI and the coregistered CT scan at the time of presentation. For clinical target volume (CTV), a margin of 3 mm was added to account for microscopic tumor spread while respecting adjacent anatomical borders. Planning target volume (PTV) was obtained by adding a margin of 3 mm. A cumulative dose of 45 Gy was prescribed to the median dose of the CTV (D50%) and applied in 5 fractions per week (single dose 1.8 Gy). In particular, patients who underwent treatment in an additive setting received the prescription dose in a composite plan, with a total of 36 Gy delivered to the tumor bed (initial tumor extension adapted to anatomic barriers) and a subsequent boost to macroscopic residues with 9 Gy. Dose constraints to normal tissue were prescribed according to the database of Quantitative Analyses of Normal Tissue Effects in the Clinic (QUANTEC) [21]. Treatment planning was performed using the treatment planning system (TPS) Siemens Syngo PT Planning software (Siemens Healthineers, Erlangen, Germany) for a patient treated before 2020, while patients treated from 2020 onwards were planned using RayStation TPS (version 8B to 11B, Raysearch Laboratories AB, Stockholm, Sweden). Proton relative biological effectiveness (RBE)-weighted dose calculation was performed assuming a constant RBE factor of 1.1, and calculated RBE-weighted doses are expressed in Gray (Gy). 

PRT treatment was delivered with active raster scanning technique under daily image-guidance. Treatments were planned either for delivery at the isocentric gantry or the fixed beamline based on the arrangement between organs at risk (OAR) and target volumes, using two to four treatment beams.

### 2.3. Follow-Up and Outcome Analysis

For assessment of tumor size reduction following PRT, the respective GTV of the original treatment plan, as well as retrospectively delineated residual tumor mass on follow-up MRI with the best tumor response, were considered. The presence of a complete remission (CR) was jointly assessed by an experienced senior radiation oncologist. Patients were excluded from the quantification of tumor size reduction when only descriptive information during radiological follow-up was available but were included in the radiological follow-up analysis for assessment of local control (LC).

The percentage of tumor volume reduction using radiotherapy was calculated using the following formula:(1)$$\mathrm{Volume}\;\mathrm{reduction}(\%)=100\;\frac{GTV-V_{post-RT}}{GTV}$$
with V_post-RT_ being the volume of potential residual tumor mass at best response following radiotherapy.

LC was calculated from the start of radiotherapy until the first diagnosis of local progression or censorship. Overall survival (OS) was calculated from the start of radiotherapy until the date of death or censorship.

### 2.4. Photon Treatment Planning and Comparative Evaluation of Treatment Plans

To assess the potential dosimetric benefits of PRT for JNA compared to commonly used conformal XRT, helical XRT plans for each patient were created retrospectively. For this purpose, the original planning CT data sets with OAR contours from the clinically approved proton plans were used. Furthermore, additional cerebral substructures for neuronal functions were contoured retrospectively. The infratentorial brain comprises the entire brainstem and cerebellum. Specifically, contouring of hippocampi and segmentation of the cerebellum into anterior and posterior lobe was performed according to previously published guidelines and landmarks, including the European Particle Therapy Network (EPTN) consensus-based atlas for CT- and MR-based contouring in neurooncology [22,23,24]. Planning of photon plans was conducted using RayStation TPS (version 11B, Raysearch Laboratories AB, Stockholm, Sweden). Radiation plans were designed according to guidelines of the authors’ institution, optimized for each patient, and reviewed by an experienced senior radiation oncologist. 

Different metrics were used to quantitatively and qualitatively evaluate the potential benefits of PRT plans compared to conformal XRT plans. For this purpose, dose volume histograms (DVH) of the different volumes were extracted from the TPS for analysis and computation in MATLAB (The MathWorks Inc., Natick, MA, USA).

The following dosimetric parameters were utilized for evaluation of the CTV: D2%, D5%, D95%, and D98% (minimum dose received in 2%, 5%, 95%, or 98% of the CTV, respectively), Dmean (mean CTV dose), V95%, V105%, V107% (volume receiving at least 95%, 105% or 107% of the prescribed dose, respectively), conformity index (CI) and homogeneity index (HI) [25].
(2)$$\mathrm{HI}=100\;\frac{D_{5\%}-D_{95\%}}{D_{p}}$$
with D_p_, the prescribed dose. The closer the HI value is to zero, the more homogenous is the dose in the CTV.
(3)$$\mathrm{CI}=100\;\frac{V_{95\%}{\left(CTV\right)}^{2}}{CTV\cdot V_{95\%}\left(Body\right)}$$
with V95% (CTV) and V95% (Body), the volume of the CTV and the body receiving at least 95% of the prescribed dose, respectively. The closer the CI is to 1, the more conformal the treatment is with respect to the normal tissue. 

Several OARs and additional OAR subvolumes were used for dosimetric evaluation and are reported in Appendix A. To assign the laterality of paired organs, we determined the principal tumor area irrespective of bilateral tumor growth. Different dosimetric parameters were extracted depending on the OAR, including Dx (minimum dose received in a portion x of the OAR volume), Dmean, or Vx (volume of the OAR receiving at least x Gy). 

The integral dose (ID), an indicator of the total energy received by an OAR during irradiation [26], is calculated as follows:(4)$$\mathrm{ID}=100\;D_{mean}\cdot V$$
with V being the volume of the respective OAR.

To account for potential complications following radiotherapy, normal tissue complication probabilities (NTCP) for various OAR were computed using different published models in the literature. These included effects on neurocognition (change in estimated intellectual quotient (IQ) [27,28] or delayed recall on the Wechsler Memory Scale-III Word List [29]), neuroendocrine dysfunctions (e.g., adrenocorticotropic or growth hormone deficiency, central hypothyroidism [30,31]), CNS necrosis [28,31,32], hearing loss or tinnitus [31], vision impairment [33,34], alopecia or erythema [35] or xerostomia [36]. Detailed information on the respective NTCP model for each OAR can be found in Table A1.

Ultimately, estimation of a potentially increased risk of developing a radiation-induced CNS malignancy following XRT compared to PRT was performed using the concept of Risk Ratio (RR). The RR represents the ratio of the excess absolute risk for secondary CNS malignancies caused by radiation with photons and protons [37,38]. RR calculation is detailed in Appendix B. 

Absolute and relative differences were calculated for evaluation of the different investigated metrics between PRT and XRT plans.

### 2.5. Statistical Analysis

Statistical analysis was carried out in MATLAB (The MathWorks Inc., Natick, MA, USA). The paired differences between dosimetric parameters, NTCP, or RR for PRT and XRT plans were assessed with a two-sided Wilcoxon signed-rank test. A *p* value < 0.05 was considered statistically significant.

## 3. Results

### 3.1. Patient and Treatment Characteristics

A total of 10 consecutive male patients were identified and analyzed in the present study. Patient and tumor characteristics are summarized in Table 1. Median age was 14 (range 12–21) years. Patients were classified according to the most common staging systems used in the literature [39]. According to Radkowski staging, all the patients were classified stage III (40% IIIa, 60% IIIb). However, one patient presented only minimal intracranial extent with the involvement of one of the skull base foramina. According to Andrews-Fisch staging, this patient was classified stage IIIc, while for the rest stage IV applied (30% IVa, 60% IVb).

For all but one patient, histological confirmation of JNA was available prior to PRT. Treatment approaches prior to PRT comprised solely medical treatment as a primary treatment after histological sampling (*n* = 1) and, most importantly, at least one (partial) tumor resection (*n* = 8). Surgical tumor resection was always preceded by tumor embolization. In seven patients, two surgeries have been performed prior to irradiation, with the second resection being performed either within 14 days after the initial surgery (*n* = 3) or in the case of a local recurrence (*n* = 4). Two patients underwent at least one further surgical intervention in the resection area due to surgery complications consisting of intracranial and subcutaneous abscesses, subcutaneous and muscular collection of pus with wound dehiscence or rhinoliquorrhoe.

At presentation, none of the patients had a history of prior irradiation. The median time from the initial diagnosis of JNA to the beginning of radiotherapy was 12.4 (range 2.3–78.0) months. PRT was performed in a definitive setting for five patients, either at initial diagnosis due to inoperability in an advanced stage (*n* = 2) or in the case of a local recurrence (*n* = 3). In all other patients, radiotherapy was performed as an additive treatment after incomplete tumor resection. One patient continued simultaneous systemic medical treatment (Sirolimus) for two weeks after the beginning of radiotherapy.

Three patients received treatment at the isocentric gantry using four beams, while all other patients were treated in the fixed beamline rooms. Among patients being treated with two beams (*n* = 3), one patient was treated using two ipsilateral beams. The remaining four patients were treated with three beams, which included a quasi-opposite beam configuration with either an additional ipsilateral beam or a quasi-craniocaudal beam at an angle of ~60° (*n* = 2) or ~105° (*n* = 2), respectively, to the ipsilateral beam.

### 3.2. Treatment Outcomes

Median radiological follow-up was 18.1 (IQR 10.5–51.5) months after the beginning of RT. Median clinical follow-up was 27.0 (IQR 13.3–58.0) months. Radiological and clinical follow-up of >40 months was available in 40% of patients. OS was 100%, and at the last follow-up, LC was 100%. Median GTV was 34.5 (range 7.0–166.0) cm^3^. During follow-up, a mean reduction in tumor volume of 87.4% was noted, including five patients showing radiological CR. 

At presentation, most common symptoms included epistaxis (*n* = 8), dys-or anosmia (*n* = 4), nasal obstruction (*n* = 3), oculoparesis (*n* = 1), visual impairment (*n* = 2), exophtalmus (*n* = 2), and rhinorrhoea (*n* = 3) of which one patient with reported pansinusitis, conductive hearing impairment due to compression of the Eustachian tube with subsequent tympanic effusion (*n* = 1), dysgeusia (*n* = 1) and facial swelling (*n* = 1). 

Most patients experienced complete or partial symptom relief during follow-up, notably regarding epistaxis and nasal obstruction. Of the patients presenting with recurrent epistaxis before radiotherapy, all but one patient were noted with only a few small bleedings, mostly during the early period following PRT. However, one patient was reported having recurrent, spontaneously ceasing, partly hemoglobin-relevant bleedings leading to hospitalizations. In consequence, three interventions to embolize tumor-supplying vessels have been performed, and treatment with Sirolimus, along with temporary treatment with tranexamic acid, have been initiated. Under currently ongoing therapy with Sirolimus, the patient had no more bleeding. During the period of follow-up, pre-existing neurological deficits such as visual impairment or oculoparesis did not significantly improve following radiotherapy. 

Acute radiation-induced toxicities included predominantly temporary mucositis, erythema, and focal alopecia. Mainly, Common Terminology Criteria for Adverse Events (CTCAE) I toxicities have been noted, with only one patient experiencing a rapidly resolving mucositis CTCAE II. No high-grade acute toxicities (>CTCAE II) have been observed. The patient receiving simultaneous systemic treatment with Sirolimus for the first two weeks of radiotherapy did not experience any increased toxicities. 

Regarding late-term side-effects, one patient developed rhinoliquorrhoe due to a skull base dehiscence approximately three years after completion of PRT. This patient has had two surgical tumor resections involving the skull base, with a subsequent first surgical revision due to rhinoliquorrhoe approximately three months prior to the initiation of PRT. For another patient, an incidental detection of a cavernoma was noted through an MRI scan conducted 3.1 years after the completion of radiotherapy. The cavernoma was found within the irradiated area, having received up to 9 Gy. One patient reported a subjective decrease in ipsilateral visual acuity, which has not been objectified by an ophthalmologist. Another patient described ipsilateral eye dryness since radiotherapy treatment. In general, objective evaluation of endocrinological and optical status could not adequately be addressed for all patients due to inconsistent check-ups and missing data. Among the patients who had undergone a complete endocrinological assessment, one patient was diagnosed with central hypothyroidism. During the follow-up period, no occurrences of secondary neoplasms have been reported.

### 3.3. Treatment Plan Comparisons

#### 3.3.1. CTV Coverage

Retrospective planning of helical XRT plans was performed, and all photon plans met the initial objectives and constraints for target coverage, and OAR was initially used for PRT. Target volume coverage was comparable for both treatment modalities. While Table 2 contains a subset of dosimetric parameters regarding CTV, all dosimetric values can be found in Appendix A.

#### 3.3.2. Sparing of OAR

While all dosimetric values related to OAR can be found in Appendix A, Table 3 specifically focuses on the most significant differences in dosimetric parameters. Significant sparing of OAR through the use of PRT included critical structures linked to the brain and its substructures, the skin, and the contralateral cochlea. Selected significant differences included an approximate 60% mean dose reduction for the contralateral cochlea. Regarding the skin, both V10 Gy and Dmean are reduced by approximatively 50%. With respect to the brain (excluding the CTV), Dmean and V10 Gy were reduced by 40% and 30%, respectively. Among the brain substructures, the infratentorial brain showed the most substantial dose reduction in PRT plans, particularly in the cerebellum, with V10 Gy close to 0% compared to 46% in XRT plans. The hippocampus is another structure where significant dose reductions are observed in the proton plans, with a mean dose reduction of 38% in the ipsilateral hippocampus and 70% in the contralateral hippocampus. 

Figure 1 shows dose distributions for the target volume and selected ROIs for a representative case of JNA treated by PRT at the isocentric gantry compared to a helical XRT plan. DVHs demonstrate similar coverage for CTV and comparable maximum doses for the optic pathways. In this specific case, the proton plan exhibits increased low-dose contributions to the eyes compared to the XRT plan. Nevertheless, notable dose reduction is observed in the supratentorial brain and, to an even greater extent, in the infratentorial brain when using PRT.

The absolute differences in NTCP values (∆NTCP), IQ (∆IQ), and RR for secondary CNS malignancies are illustrated in Figure 2 and detailed in Appendix A. A subset of those values is presented in Table 4. Significant differences in NTCP values were most importantly related to the brain, the skin, and the auditory system. Less dose deposition in bilateral hippocampi and infratentorial brain in proton plans translated with significantly improved estimates for delayed recall (∆NTCP of 20%) and IQ (∆IQ of ~5 points), respectively. Favourable dose distribution to skin and the auditory system predicted better outcomes for alopecia (∆NTCP of ~7%) and tinnitus on the contralateral ear (∆NTCP of ~3%). The average estimated risk of radiation-induced secondary CNS malignancies was about two times higher for photon plans compared to proton plans, and in certain scenarios, it increased up to five times higher.

## 4. Discussion

Advanced stages of JNA with intracranial involvement present a therapeutic challenge due to their close proximity to critical structures such as cranial nerves at the skull base or endocrinological structures. In incompletely resected JNA or tumors that are deemed irresectable without major morbidity, radiotherapy is known to be an effective treatment approach [6,7]. Given the benign nature of JNA and the young age of the affected patients, a certain reluctance regarding this treatment approach consists of long-term radiation-induced morbidity and life quality. This legitimate concern could only partly be allayed by the introduction of conformal XRT [7,40]. 

PRT represents a promising modality for the treatment of benign conditions in young patients, considering its high degree of target conformity together with its steep dose gradient and low integral dose in normal tissue, thus bearing the potential to reduce long-term sequelae [19]. 

In this single-center study, we retrospectively analyzed the outcome of patients with advanced JNA treated with PRT at the Heidelberg Ion Beam Therapy Center and, in a further step, evaluated the dosimetric benefits of PRT compared to conformal XRT. 

In our patient cohort, PRT yielded an excellent LC rate of 100% at the last follow-up with a dose prescription of 45 Gy in 25 fractions, which was partially applied with the boost concept. Previously, LC rates of 73% to 100% with prescription doses ranging from 30 Gy to 55 Gy have been reported [4,6,15]. To date, the optimal dose regimen is not conclusively established for JNA. Some authors have described a dose relation for tumor control in JNA. For instance, in a retrospective study of 15 patients, tumor recurrence within two years of treatment was reported in four of five patients receiving 32 Gy at 2 Gy, while patients receiving 36–46 Gy at 2 Gy did not show any signs of tumor recurrence [41]. Amdur et al. described an LC rate of 77% with a prescription dose of 30 Gy at 1.43 Gy with all recurrences occurring in the field, while LC amounted to >90% when using a higher dose regimen consisting of 35–36 Gy at 1.7–1.8 Gy [42]. Similarly, Mc Afee et al. noted in a retrospective study of 22 patients that all of the recurrences occurred in patients who were prescribed 30–31.8 Gy in 18 to 22 fractions, while patients receiving at least 35 Gy remained controlled [43]. Advances in radiation therapy techniques, such as the implementation of conformal radiotherapy (IMRT, VMAT, helical therapy), have enabled the delivery of radiation with enhanced precision, thus allowing high doses to the tumor to ensure optimal tumor control while potentially minimizing morbidity. This holds particular importance when addressing advanced cases of JNA, as they typically manifest in close proximity to critical structures. The limited existing reports focusing on IMRT for JNA have outlined its effectiveness in achieving LC while providing superior dosimetry for critical structures [4,40]. A retrospective study conducted by Chakraborty et al. focusing on conformal radiotherapy for JNA featured the largest cohort treated with IMRT to date. With 7 out of 8 patients receiving IMRT with a median prescription dose of 39.6 Gy, a 2-year tumor control rate of 87.5% was achieved [4]. Given its superior degree of conformality due to its advantageous physical properties, PRT allowed safe dose application of 45 Gy to the tumor in all of our patients presenting with advanced-staged JNA Radkowski III with partly extensive intracranial tumor spread. Potentially, this consistently higher dose regimen may additionally contribute to the excellent local control of 100% observed in our small patient cohort. Another potential contributing factor may rely on a possible higher biological efficiency of protons for tissues with a low α/β ratio that may lead to an RBE value superior to the constant value of 1.1 used in clinical routine [44].

PRT led to considerable tumor shrinkage. During follow-up, we observed a mean reduction in tumor volume of 87.4%, including five patients achieving a CR, occurring primarily in the first 2 years. Tumor volume reduction led to an improvement in conductive hearing impairment due to decompression of the Eustachian tube and partly translated into an improvement in nasal obstruction during follow-up. However, no significant impact on pre-existing neurological deficits such as visual impairment or oculoparesis have been observed in the two patients concerned, possibly due to neuronal damage lasting too long before decompression.

In line with previous reports regarding conformal radiotherapy for JNA, radiotherapy was well tolerated, with acute side-effects comprising only low-grade acute side-effects (CTCAE I–II) with predominantly temporary skin erythema and mucositis [4,6,40].

In literature, various long-term radiation-induced sequelae have been reported, such as xerostomia with subsequent caries, cataracts, panhypopituitarism, and growth retardation or secondary neoplasms [6,15,42,45]. In our cohort, one patient experienced rhinoliquorrhoe approximately three years after PRT, requiring an extensive revision of the skull base. It has to be noted that this patient presented with a prior history of two partial tumor resections, followed by surgical reconstruction of the skull base and another revision surgery, including mesh insertion after an initial episode of rhinoliquorrhoe before the initiation of adjuvant radiotherapy for residual tumor. Given that the target volume comprised the surgically revised area and with regard to the CR noted during follow-up, a contribution of radiotherapy to the second episode of rhinoliquorrhoe cannot be ruled out. Indeed, tumor shrinkage has been considered a possible cause of late-onset cerebrospinal fluid (CSF) leakage after antineoplastic therapy for other tumor entities with invasion of the skull base, such as in the case of advanced pituitary macroadenoma [46,47]. Likewise, CSF leakage induced using the flutamide treatment has already been described as a rare complication associated with post-therapeutic tumor shrinkage in a case report of a young patient with advanced JNA extending into the anterior skull base [48]. 

In the case of a 14-year-old patient, an MRI conducted approximately 3 years after completion of radiotherapy incidentally indicated a potential cavernoma, which was found within the irradiated area having received up to 9 Gy. Based on a neurological assessment, it was concluded that continuous monitoring through regular follow-up was the appropriate course of action, and no further intervention was deemed necessary.

Radiation-induced cavernomas (RIC) have been identified as common long-term side-effects in patients having received radiotherapy at a young age [49,50]. In a retrospective comparative study between PRT and XRT in the treatment of pediatric medulloblastoma, Trybula et al. noted that cavernous malformations are detected significantly earlier in patients having undergone proton beam therapy [51]. Nevertheless, there was no observed difference in the occurrence of RIC that required intervention beyond regular follow-up. Long-term follow-up of childhood cancer survivors showed a low risk of symptomatic hemorrhages due to RIC, with the majority of cases having a complication-free course and not needing intervention [50]. To date, there are no documented cases of RIC specifically related to JNA. However, many prior studies reporting outcomes of JNA treatment using radiotherapy regularly rely on follow-up imaging conducted through CT rather than MRI, which is, however, the modality of choice for the detection of cavernomas [4,6,52,53]. 

One patient was diagnosed with central hypothyroidism after radiotherapy, with the treatment target volume encompassing the pituitary in the majority of our patients. However, objective evaluation of long-term side effects, including endocrinological and optical status, could not be adequately addressed for all patients due to inconsistent check-ups and missing data.

In order to better assess the potential benefits of PRT for young patients with JNA, we conducted comprehensive comparisons aimed at evaluating the dosimetric advantages and potential reductions in radiation-induced complications between PRT and XRT plans. 

Regarding CTV coverage, HI and CI were comparable between the two treatment modalities, similar to other studies showing only marginal gain by the use of PRT at higher prescribed dose levels [54,55]. Indeed, in this study, most OARs have maximum dose constraints set above the prescription dose, thus limiting the potential benefit of the favorable dose distributions offered by PRT regarding CTV coverage.

The primary dosimetric advantages of PRT over helical XRT were observed for the contralateral cochlea, for the skin, and, notably, for the brain and its substructures. Sparing of the pituitary was not feasible for seven patients due to its inclusion in the target volume. Nevertheless, for two patients with partial inclusion of the pituitary in the target volume and one patient with the pituitary located at the border of the treated volume, a substantial reduction in the mean dose of 10–40% could be achieved. Dose distribution to the supratentorial brain and hippocampi remained substantially reduced even in cases with a proton beam being directed through the supratentorial part of the brain. Previous studies have associated radiation of the hippocampus, in particular, with potential learning and memory impairment [56]. Ensuring minimal doses to these critical structures is of paramount importance, particularly in light of previous findings that highlighted negative effects on IQ when supratentorial volumes received doses as low as 0–5 Gy [57]. More importantly, PRT substantially spared the infratentorial part of the brain, particularly the posterior lobe of the cerebellum. Growing evidence suggests that the cerebellum, specifically the posterior lobe of the cerebellum, plays a crucial role in various complex cognitive functions. This is further supported by the observation that irradiation of both cerebellar lobes is associated with a decline in IQ, while a decline in academic achievement scores is primarily attributed to the mean posterior cerebellar dose [58,59]. Our dosimetric results indicating the possibility of improved neurocognitive outcomes with PRT were reinforced through comparisons of NTCP and IQ between the two treatment modalities. The more pronounced delayed recall and decline in IQ predicted for helical XRT, attributed to increased dose deposition in the hippocampi and infratentorial brain, underscores the potential of PRT in mitigating cognitive sequelae by sparing those critical brain regions more effectively, thus potentially enhancing the patients’ quality of life.

Consistent with the observation that PRT generally allows better sparing of contralateral organs [54], our study found a reduced risk of tinnitus in the contralateral cochlea following PRT. However, this effect was moderate compared to findings for other indications reported in the literature, possibly due to the relatively low prescribed dose of 45 Gy used in our study. NTCP for ipsilateral OARs showed varying differences depending on the target volume characteristics and specific beam arrangements for each patient. 

Additionally, our research yielded encouraging findings regarding the potential of PRT for patients with JNA in terms of reducing the excess absolute risk for secondary CNS malignancies compared to XRT. The significant reduction in integral dose to the brain, with notably reduced low doses to the infratentorial brain, indicates a potential risk reduction by the use of PRT, as it has been previously observed that low-to-intermediate radiation doses can increase the risk for radiation-induced secondary malignancies in children [60]. Subsequent risk assessments revealed an average risk reduction of 52%. The extent of this risk reduction depended on the specific beam arrangement employed, with gantry-based PRT yielding the highest risk reduction, exhibiting RR values around two or higher and even reaching up to approximately five in some cases. 

These findings underscore the significance of individual patient factors and treatment setups in determining the potential benefits of PRT in terms of specific long-term complications [37]. Individualized treatment planning is vital in maximizing the advantages of PRT, striking the right balance between minimizing potential acute toxicity and reducing long-term risks associated with radiation exposure. Using the threshold defined by Dutz et al., which considers that any ∆NTCP > 10% or higher in favor of PRT would warrant selecting PRT over XRT, all patients in our cohort would have been candidates for PRT [54].

Putting our results into perspective requires careful consideration of certain factors. A potential confounding factor is the consequent use of MRI for treatment planning and follow-up in all of our patients, while several studies on conformal radiotherapy for JNA partly rely on CT scans as the preferred imaging modality [4,6]. However, MRI imaging offers distinct advantages in discrimination between tumors and inflamed mucosa in the nasal cavity and paranasal sinuses, which often accompanies the disease. Specifically, for tumors that are typically located at the skull base with a propensity for intracranial extent, MRI provides superior differentiating and precise delineating of the tumor and surrounding OAR. Besides conformal radiation technologies, MRI imaging thus allows less expanded target volumes, ultimately reducing side-effects in surrounding tissue while ensuring effective tumor control. Furthermore, MRI is considerably more sensitive in detecting tumor regression and recurrences as well as possible long-term complications during follow-up assessments. In terms of methodology, the retrospective study design introduces the potential for additional confounding factors and inconsistencies in data collection. Another major limitation of this study is the limited number of patients in our cohort. This is similar to most previous studies on radiotherapy for JNA and inevitably due to its rare nature. In 1984, Cummings et al. reported on the largest cohort to date, which included 55 patients treated over a period of two decades [45]. However, the use of conventional radiotherapy, together with imaging and planning approaches prevalent at that time, made it challenging to evaluate and compare the results effectively. Not least to the relatively short follow-up of our study, the results need to be evaluated with caution. Nevertheless, the conducted comparisons of NTCP and risk assessments for secondary neoplasms can provide valuable insights into the potential reduction in side effects when utilizing PRT in comparison to highly conformal radiotherapy techniques like helical therapy.

It is important to acknowledge that the estimation of radiation-induced impairments may also be subject to uncertainties. The utilized models were originally designed for different medical conditions and prescribed dose levels. In the case of risk assessment for secondary CNS malignancies, for instance, the respective model does not account for secondary neutrons [37], which may impact the accuracy of predictions. Despite these limitations, these models still serve as robust tools for objectively and qualitatively comparing the potential clinical impact of different treatment modalities.

Taken together, this study strongly suggests that PRT with a dose of 45 Gy is a safe and effective therapeutic option for the treatment of advanced JNA. Notably, reducing potential long-term radiation-induced complications is of utmost importance, especially considering the young age of patients with JNA and their considerable life expectancy. Further research, including long-term follow-up, is necessary to expand on these findings and further explore the advantages of PRT for treating advanced JNA.

## 5. Conclusions

To our knowledge, this is the first report of PRT for recurrent or inoperable advanced JNA, providing promising insights into the rationale of its use for clinical application in a postoperative or definitive setting. PRT with a dose of 45 Gy achieved excellent LC rates with minimal acute morbidity and the potential for reduced radiation-induced long-term complications, which is particularly

## Figures and Tables

**Figure 1 cancers-15-05022-f001:**
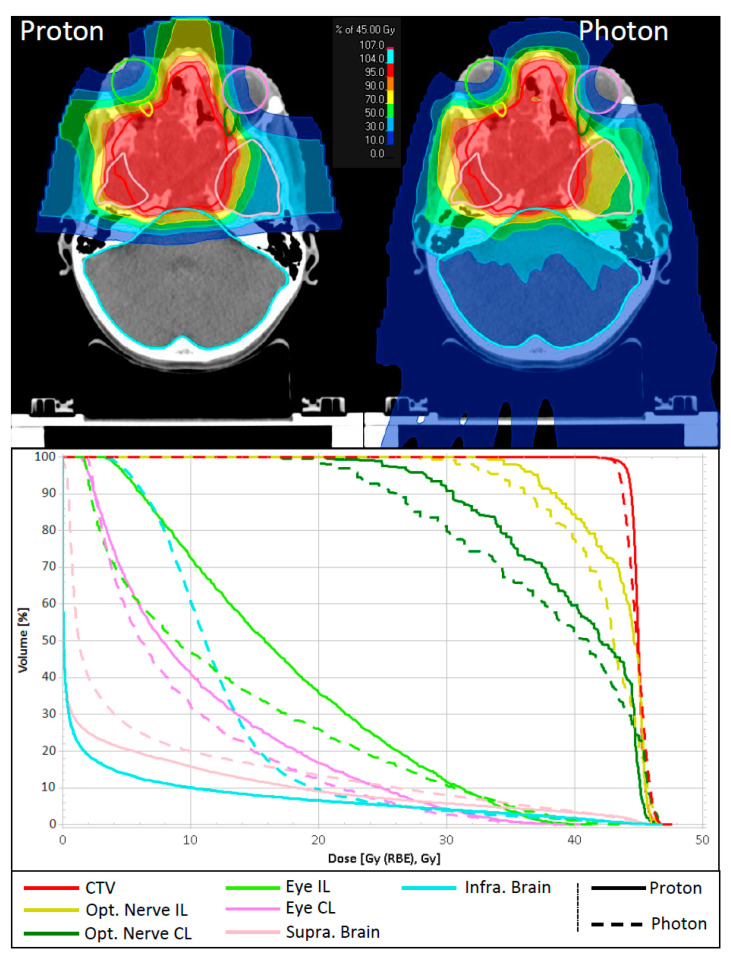
Comparison of dose distribution and dose volume histograms between a proton treatment plan and a helical photon therapy plan for a representative patient. CTV and selected OAR are shown. CTV: clinical target volume; Opt. Nerve IL/CL: ipsi- and contralateral optic nerve; Eye IL/CL: ipsi- and contralateral eye; Supra. Brain: supratentorial brain; Infra. Brain: the infratentorial brain.

**Figure 2 cancers-15-05022-f002:**
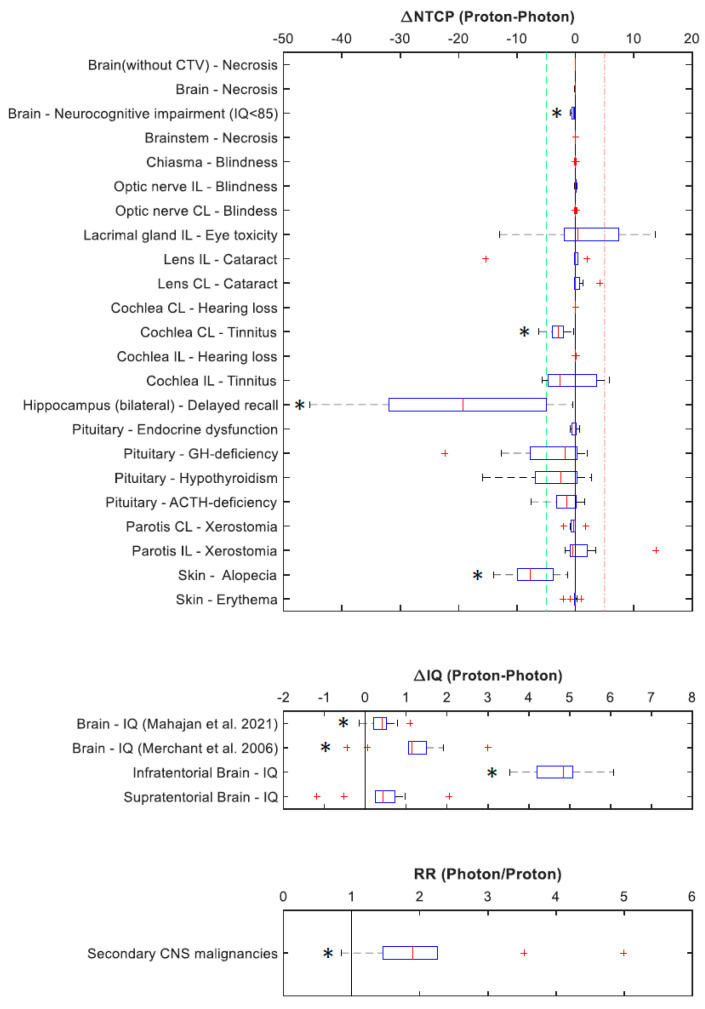
Differences in normal tissue complication probabilities (∆NTCP), differences in intellectual quotients (∆IQ), and risk ratio (RR) for secondary CNS malignancies following PRT and XRT. * indicates significant results. CL: contralateral; IL: ipsilateral; GH-deficiency: growth hormone deficiency; ACTH-deficiency: adrenocorticotropic hormone deficiency [27,28].

**Table 1 cancers-15-05022-t001:** Patient and tumor characteristics. For Patients 1 and 7, only descriptive information on follow-up MRI was available.

Patient	Age at the Time of RT	GTV (cm^3^)	Post-RT Tumor Volume (cm^3^)	Tumor Volume Reduction (%)
1	13	28	-	-
2	14	159	29	81.8
3	19	8	0	100.0
4	15	29	0	100.0
5	21	34	18	47.1
6	12	166	49	70.5
7	20	89	-	-
8	12	35	0	100.0
9	12	67	0	100.0
10	14	7	0	100.0

GTV: gross tumor volume; RT: radiotherapy; Post-RT tumor volume corresponds to the posttherapeutic tumor volume at best response; - is used as a placeholder when only descriptive information on follow-up MRI was available, which made volume reduction calculation not applicable.

**Table 2 cancers-15-05022-t002:** Dosimetric parameters regarding CTV. Doses are expressed in percentage of the prescribed dose. Volumes are expressed in percentages of the CTV volume.

		Proton			Photon			Δabs (Proton-Photon)			Δrel (Proton-Photon)			
		Mean	±	SD	Mean	±	SD	Mean	±	SD	Mean	±	SD	*p*-Value
CTV	D_0.03 cm³_	105.5	±	2.3	104.4	±	0.7	1.1	±	2.2	1.0	±	2.1	0.193
	D_2%_	102.9	±	0.7	103.1	±	0.4	−0.2	±	0.7	−0.2	±	0.7	0.263
	D_95%_	97.1	±	1.8	97.3	±	0.6	−0.2	±	0.8	−0.2	±	1.9	0.769
	V_95%_	97.8	±	4.4	99.6	±	0.5	−1.8	±	4.3	−1.8	±	4.3	0.250
	V_107%_	0.1	±	0.2	0.0	±	0.0	0.1	±	0.2	-	±	-	-
	HI	5.1	±	1.8	5.4	±	0.8	−0.3	±	1.8	−5.6	±	30.5	0.695
	CI	0.58	±	0.18	0.56	±	0.16	0.03	±	0.06	4.8	±	9.0	0.287

CTV: clinical target volume; V(x%): volume of the CTV receiving a minimum dose of x% of the prescribed dose; D_0.03 cm³_: minimum dose received in 0.03 cm^3^ of the CTV; Dx%: minimum dose received by x% of the CTV; HI: homogeneity index; CI: conformity Index; SD: standard deviation; ∆abs: absolute difference; ∆rel: relative difference in %. - is used as a placeholder when dosimetric parameters were close to zero for all treatment modalities, which made relative comparison not applicable.

**Table 3 cancers-15-05022-t003:** Selected dosimetric parameters related to organs at risk with significant differences between proton and photon plans. Doses are expressed in Gy. Integral dose (ID) is expressed in Gy × cm^3^.

		Proton			Photon			Δabs (Proton-Photon)			Δrel (Proton-Photon)			
		Mean	±	SD	Mean	±	SD	Mean	±	SD	Mean	±	SD	*p*-Value
Brain (without CTV)	D_0.03 cm³_	45.4	±	0.9	45.9	±	0.6	−0.4	±	0.6	−1.0	±	1.4	0.006
	D_2%_	29.7	±	10.0	32.9	±	7.0	−3.2	±	3.8	−12.0	±	14.6	0.004
	D_50%_	0.1	±	0.1	1.4	±	1.3	−1.4	±	1.3	−93.3	±	12.0	0.002
	D_mean_	3.0	±	1.4	5.1	±	1.6	−2.1	±	1.7	−39.1	±	29.3	0.006
	V_10 Gy_	11.4	±	6.7	17.6	±	6.3	−6.1	±	7.9	−31.3	±	43.9	0.049
	V_15 Gy_	7.2	±	4.0	10.0	±	4.1	−2.8	±	3.0	−28.3	±	29.1	0.020
	V_20 Gy_	4.7	±	2.6	6.7	±	3.2	−2.0	±	1.4	−30.9	±	18.9	0.004
	V_35 Gy_	1.7	±	1.2	2.1	±	1.4	−0.3	±	0.2	−20.0	±	14.7	0.002
	ID	4924.2	±	2432.6	8294.5	±	2969.2	−3370.3	±	2755.4				
Supratentorial Brain	D_2%_	31.5	±	11.7	34.8	±	8.5	−3.3	±	3.9	−12.0	±	14.9	0.027
	D_50%_	0.1	±	0.1	0.9	±	0.5	−0.9	±	0.6	−88.2	±	24.0	0.002
	V_20 Gy_	5.2	±	3.1	7.1	±	3.6	−1.8	±	1.6	−27.5	±	21.7	
Infratentorial Brain	D_0.03 cm³_	43.1	±	6.8	44.1	±	5.5	−1.0	±	1.6	−2.8	±	4.9	0.035
	D_50%_	0.1	±	0.0	10.1	±	1.7	−10.0	±	1.7	−99.3	±	0.4	0.002
	D_mean_	2.8	±	1.4	11.4	±	2.1	−8.7	±	1.3	−76.7	±	8.9	0.002
	V_10 Gy_	8.4	±	4.4	52.2	±	17.2	−43.8	±	15.6	−83.6	±	6.3	0.002
	V_15 Gy_	6.4	±	3.9	17.9	±	7.7	−11.4	±	4.7	−65.2	±	12.2	0.002
	V_20 Gy_	5.1	±	3.3	7.8	±	4.9	−2.7	±	2.0	−38.6	±	21.7	0.002
	ID	679.0	±	353.5	2770.8	±	653.4	−2091.8	±	383.0				
Cerebellum	D_0.03 cm³_	13.7	±	10.6	26.3	±	7.4	−12.6	±	5.3	−53.5	±	28.6	0.002
	D_2%_	3.1	±	2.9	18.8	±	4.1	−15.8	±	2.3	−85.4	±	10.9	0.002
	D_50%_	0.0	±	0.0	9.6	±	1.6	−9.6	±	1.6	−99.7	±	0.2	0.002
	D_mean_	0.3	±	0.2	10.0	±	1.8	−9.7	±	1.6	−97.3	±	1.8	0.002
	V_10 Gy_	0.4	±	0.6	45.9	±	19.9	−45.5	±	19.6	−99.4	±	1.0	0.002
	V_15 Gy_	0.2	±	0.3	8.6	±	6.3	−8.5	±	6.1	−98.6	±	1.8	0.002
	V_20 Gy_	0.1	±	0.1	1.9	±	2.3	−1.8	±	2.3	−66.6	±	46.2	0.016
	ID	43.6	±	31.2	1522.7	±	312.4	−1479.1	±	295.7				
Cerebellum anterior	D_0.03 cm³_	9.2	±	9.3	20.8	±	6.9	−11.5	±	5.8	−61.4	±	30.1	0.002
	D_2%_	3.7	±	4.2	15.8	±	4.8	−12.1	±	3.5	−79.4	±	17.9	0.002
	D_50%_	0.1	±	0.1	7.1	±	2.5	−7.0	±	2.5	−98.9	±	0.9	0.002
	D_mean_	0.4	±	0.4	7.5	±	2.4	−7.1	±	2.2	−95.2	±	4.0	0.002
	V_10 Gy_	0.5	±	1.2	25.7	±	23.1	−25.2	±	22.7	−98.5	±	2.5	0.002
	V_15 Gy_	0.2	±	0.6	3.2	±	3.2	−2.9	±	2.9	−70.9	±	40.5	0.016
	V_20 Gy_	0.1	±	0.3	1.0	±	1.6	−0.9	±	1.4	−53.1	±	46.7	0.031
	ID	6.2	±	5.2	128.1	±	39.3	−121.9	±	38.2				
Cerebellum posterior	D_0.03 cm³_	12.1	±	9.5	26.0	±	7.4	−13.9	±	4.2	−58.5	±	25.1	0.002
	D_2%_	3.0	±	3.1	19.0	±	4.2	−16.0	±	2.2	−86.0	±	11.3	0.002
	D_50%_	0.0	±	0.0	9.9	±	1.6	−9.8	±	1.6	−99.8	±	0.2	0.002
	D_mean_	0.3	±	0.2	10.3	±	1.7	−10.0	±	1.5	−97.5	±	1.9	0.002
	V_10 Gy_	0.3	±	0.7	48.4	±	19.8	−48.0	±	19.5	−99.4	±	1.0	0.002
	V_15 Gy_	0.1	±	0.3	9.3	±	6.9	−9.2	±	6.7	−98.8	±	1.9	0.002
	V_20 Gy_	0.1	±	0.1	2.0	±	2.5	−2.0	±	2.4	−67.2	±	46.6	0.016
	ID	37.2	±	28.3	1395.7	±	285.2	−1358.6	±	271.9				
Hippocampus (bilateral)	D_0.03 cm³_	30.2	±	11.1	34.5	±	6.2	−4.3	±	5.4	−15.3	±	20.5	0.049
	D_2%_	27.7	±	11.3	32.0	±	7.0	−4.3	±	5.1	−16.8	±	21.7	0.049
	D_40%_	4.3	±	5.5	13.7	±	8.1	−9.3	±	5.1	−70.4	±	24.3	0.002
	D_50%_	2.3	±	3.6	10.3	±	6.9	−8.0	±	4.8	−80.0	±	17.1	0.002
	D_mean_	6.0	±	3.9	12.2	±	5.4	−6.2	±	3.6	−49.8	±	27.1	0.004
	ID	24.5	±	14.5	51.7	±	24.7	−27.3	±	18.3				
Hippocampus CL	D_0.03 cm³_	20.1	±	15.5	28.9	±	11.0	−8.9	±	7.5	−39.5	±	31.3	0.006
	D_2%_	19.1	±	15.0	28.2	±	11.0	−9.1	±	7.3	−41.0	±	31.1	0.006
	D_50%_	1.5	±	2.3	9.6	±	6.3	−8.1	±	4.9	−86.5	±	12.6	0.002
	D_mean_	4.0	±	4.0	11.4	±	5.6	−7.4	±	3.7	−69.6	±	20.0	0.002
	ID	7.9	±	7.3	24.1	±	13.2	−16.2	±	10.2				
Hippocampus IL	D_0.03 cm³_	29.3	±	11.8	33.4	±	6.7	−4.1	±	5.5	−15.7	±	22.4	0.049
	D_2%_	28.6	±	11.9	32.7	±	6.9	−4.1	±	5.4	−16.3	±	23.0	0.049
	D_50%_	5.5	±	6.2	11.5	±	7.6	−6.0	±	6.4	−35.0	±	112.2	0.037
	D_mean_	8.1	±	4.9	13.2	±	5.3	−5.0	±	4.1	−38.2	±	37.6	0.010
	ID	16.6	±	9.3	27.7	±	12.3	−11.1	±	9.1				
Cochlea CL	D_0.03 cm³_	8.3	±	7.0	22.1	±	4.7	−13.8	±	6.3	−63.2	±	29.0	0.002
	D_2%_	9.8	±	7.4	23.7	±	4.9	−14.0	±	6.6	−59.6	±	28.1	0.002
	D_mean_	7.5	±	6.4	21.1	±	4.5	−13.6	±	5.8	−65.3	±	28.1	0.002
	ID	0.9	±	0.8	2.6	±	1.1	−1.7	±	0.9				
Skin	D_2%_	20.9	±	6.9	23.2	±	6.2	−2.3	±	2.7	−10.7	±	12.2	0.049
	D_5%_	14.2	±	5.3	18.0	±	4.3	−3.8	±	2.0	−23.2	±	13.7	0.002
	D_mean_	2.3	±	1.1	4.4	±	1.4	−2.1	±	0.5	−50.0	±	9.1	0.002
	V_10 Gy_ *	52.4	±	30.9	95.8	±	44.1	−43.3	±	17.8	−47.4	±	12.3	0.002
	V_15 Gy_ *	29.5	±	27.0	45.3	±	27.9	−15.9	±	7.4	−42.3	±	18.6	0.002
	V_20 Gy_ *	14.2	±	13.0	21.0	±	16.6	−6.8	±	6.5	−28.7	±	50.4	0.004
	ID	1294.0	±	625.4	2498.2	±	876.4	−1204.3	±	311.8				

Volumes are expressed in % of the structure volume, except for the skin volumes (*), which are expressed in cm³. VxGy: volume of the OAR receiving a minimum dose of × Gy; D_0.03 cm³_: minimum dose received in 0.03 cm^3^ of the OAR; Dx%: minimum dose received by x% of the OAR. CTV: clinical target volume; CL: contralateral; IL: ipsilateral, SD: standard deviation; ∆abs: absolute difference; ∆rel: relative difference in %.

**Table 4 cancers-15-05022-t004:** Estimated intelligence quotient (IQ) following radiotherapy, normal tissue complication probabilities (NTCP) for selected organs at risk, and risk ratio (RR) for secondary CNS malignancies. NTCP is expressed in %, and IQ (*) is expressed in points.

		Proton			Photon			Δabs (Proton-Photon)			
Organ at Risk	Complication	Mean	±	SD	Mean	±	SD	Mean	±	SD	*p*-Value
Infratentorial Brain	IQ *	112.4	±	4.1	107.6	±	4.3	4.7	±	0.7	0.002
Cochlea CL	Tinnitus	0.8	±	0.7	3.7	±	2.1	−2.9	±	1.8	0.002
Cochlea IL	Tinnitus	7.6	±	7.7	8.8	±	4.6	−1.1	±	4.2	0.432
Hippocampus (bilateral)	Delayed recall	8.5	±	10.7	28.3	±	24.1	−19.9	±	15.6	0.002
Pituitary	GH-deficiency	62.6	±	20.1	67.2	±	13.1	−4.6	±	7.7	0.131
Pituitary	Hypothyroidism	43.9	±	18.6	47.4	±	13.7	−3.5	±	5.6	0.131
Pituitary	ACTH-deficiency	25.0	±	9.2	26.7	±	6.9	−1.7	±	2.7	0.131
Lacrimal gland IL	Ocular toxicity	18.7	±	28.4	17.2	±	24.7	1.5	±	7.5	0.557
Parotis CL	Xerostomia	2.2	±	1.6	2.5	±	1.0	−0.3	±	0.9	0.084
Parotis IL	Xerostomia	5.9	±	7.9	4.4	±	3.8	1.5	±	4.6	0.695
Skin	Alopecia	45.3	±	10.3	52.7	±	8.4	−7.4	±	3.9	0.002
		**RR (Photon/Proton)**									
		Mean	±	SD	*p*-Value						
Brain	Secondary malignancies	2.1	±	1.2	0.006						

CL: contralateral; IL: ipsilateral, SD: standard deviation. GH-deficiency: growth hormone deficiency; HT: hypothyroidism; ACTH-deficiency: adrenocorticotropic hormone deficiency; RR: risk ratio.

## Data Availability

Research data are stored in an institutional repository and will be shared upon request to the corresponding author.

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
