# Peer review of "Proton Therapy for Advanced Juvenile Nasopharyngeal Angiofibroma"

_cancers, 2023, doi:10.3390/cancers15205022_

Round 1
Reviewer 1 Report
This is a well-written and very well-documented paper about advance stage inoperable or recurrent JNA, where surgery is not the tool.
Major concern: reading the paper authors may have the feeling that previously conventional irradiation was the primary tool in this disease and not surgery, but in reality always embolization and surgery is the first consideration. Therefore I would suggest to add to the Title of the paper "inoperable" advance stage and "recurrent" JNA to make it more clear that irradiation is only the second choice in these cases, where the proton therapy is much better than conventional radiation techniques
Minor comment:
Line 109-112: "3. Results This section may be divided by subheadings. It should provide a concise and precise 110 description of the experimental results, their interpretation, as well as the experimental 111 conclusions that can be drawn." should be omitted
Line 383: INSTITUTION-XXX should specified
All abbreviation should be resolved at first use (e.g. EPTN)
Reviewer 2 Report
This is a report of proton therapy for advanced juvenile nasopharyngeal angiofibroma. There have been no reports of proton therapy for this tumor, making it novel.
Since this tumor tends to occur at a young age and is benign, the ability to treat it without quality of life deterioration and late adverse events such as second cancers due to radiation therapy is a serious concern. The authors have shown that proton therapy can reliably lower doses to risk organs such as the central nervous system compared to high-precision photon therapy, which is certainly advantageous in reducing the likelihood of late adverse events. I agree with the authors that proton therapy is useful because the patients are young and low-dose exposures should be avoided if possible. Prediction of late adverse events by irradiated dose has also been examined in detail.
The limitations of this study are also pertinent. I look forward to further reports on how the actual late effects compare to the predicted values.
Author Response
This is a report of proton therapy for advanced juvenile nasopharyngeal angiofibroma. There have been no reports of proton therapy for this tumor, making it novel.
Since this tumor tends to occur at a young age and is benign, the ability to treat it without quality of life deterioration and late adverse events such as second cancers due to radiation therapy is a serious concern. The authors have shown that proton therapy can reliably lower doses to risk organs such as the central nervous system compared to high-precision photon therapy, which is certainly advantageous in reducing the likelihood of late adverse events. I agree with the authors that proton therapy is useful because the patients are young and low-dose exposures should be avoided if possible. Prediction of late adverse events by irradiated dose has also been examined in detail.
The limitations of this study are also pertinent. I look forward to further reports on how the actual late effects compare to the predicted values.
We would like to thank the reviewer for this great feedback.
Reviewer 3 Report
The results presented in the manuscript demonstrate new data on the use of proton therapy in the treatment of young patients.
Good quality article. Can be accepted in this form
Author Response
The results presented in the manuscript demonstrate new data on the use of proton therapy in the treatment of young patients.
Good quality article. Can be accepted in this form
We would like to thank the reviewer for this great feedback.
Reviewer 4 Report
In their original report, Hoeltgen and colleagues meticulously analyze the treatment outcomes of 10 consecutive patients with juvenile nasopharyngeal angiofibroma (JNA) who underwent proton therapy (PRT). Despite being classified as a rare and benign tumor, JNA's precarious location poses a significant threat to vital structures in the skull base, including the brain. The authors adeptly outline the challenges in diagnostics and explore various treatment options, delving into the associated risks such as secondary neoplasms or neuronal deficits resulting from radiation therapies.
Remarkably, this study appears to be groundbreaking as the first investigation into the application of PRT in JNA patients. The authors skillfully navigate through the information and limitations of the small study group, providing a well-balanced assessment. Consequently, they assert that PRT emerges as a (relatively) safe alternative, advocating for its preference over conventional radiotherapies.
Minor issue: Line 383: INSTITUTION-XXX should be corrected in the text
